# Crash Response of Laser-Welded Energy Absorbers Made of Docol 1000DP and Docol 1200M Steels

**DOI:** 10.3390/ma14112808

**Published:** 2021-05-25

**Authors:** Paweł Prochenka, Jacek Janiszewski, Michał Kucewicz

**Affiliations:** 1Faculty of Mechatronics, Armament and Aviation, Institute of Armament Technology, Military Univesity of Technology, 2 Gen. S. Kaliskiego Street, 00-908 Warsaw, Poland; jacek.janiszewski@wat.edu.pl; 2Faculty of Mechanical Engineering, Institute of Mechanics and Applied Computer Science, Military University of Technology, 2 Gen. S. Kaliskiego Street, 00-908 Warsaw, Poland; michal.kucewicz@wat.edu.pl

**Keywords:** advanced high-strength steels, laser welding, split Hopkinson pressure bar, crashworthiness

## Abstract

The crushing response of a laser-welded square tube absorber made of two commercial steel grades, Docol 1000DP and Docol 1200M, is presented in the paper. Crush experiments are performed at two different loading conditions, namely, quasi-static loading at 0.5 mm/s deformation speed and impact loading at 25–28 m/s. A new approach has been proposed to study the square tube absorber under impact loading using a direct impact Hopkinson (DIH) method. To characterize the mechanical properties of the tested steels, tensile quasi-static and high strain rate testing are also performed with the use of specimens with a 7 mm gauge length. The applied strain rates are 10^−3^, 10^0^, and above 10^3^ s^−1^. The laser-welded joints are also characterized by microhardness test involving the base material, heat-affected zone, and fusion zone. The crashworthiness of model square tube absorbers is estimated based on the following parameters: absorbed energy, mean force, crushing force efficiency factor, and specific energy absorbed. It has been found that the square tube absorbers made of Docol 1200M steel show a higher potential in mechanical energy absorption capacity than Docol 1000DP absorber. Moreover, crushing tests prove that laser-welded joints in 0.6 mm sheets made of Docol 1000DP and Docol 1200M steels reveal high cracking toughness. In turn, strength testing at different strain rates confirms the higher strain rate sensitivity of Docol 1000DP steel than in the case of Docol 1200M steel as well as an increase in the high ductility properties of both steel grades under the high strain rate loading conditions.

## 1. Introduction

Currently, due to environmental issues, the automotive industry is focused on the development of fuel-efficient and electric vehicles [1,2,3,4]. In both cases, engineers confront new challenges, of which the most important is to decrease the vehicle mass while increasing safety to prevent passenger injuries and, in electric cars, to protect sensitive components such as batteries. This can be achieved in two different ways: application of novel complex energy-absorbing structures with various shapes, composition, and filling [5,6,7] or the use of advanced materials with increased strength [8,9,10,11,12,13]. Therefore, composites and high-strength alloys have gained popularity in the past few decades [14,15]. The limited application of composites (mostly carbon fibers) is caused by the relatively high costs of production and is mostly present in luxury and high-end vehicles [16]. Common cars whose production costs are intended to be minimal adopt advanced high-strength steels (AHSS), high-strength aluminum alloy (HSAA) [17], and magnesium alloys for manufacturing the passive safety components. These materials, due to their mechanical properties, crashworthiness, and recycling potential [18], belong to a group of materials with a high development potential, which is exemplified by the introduction of the third generation of AHSS to the market.

The effective design of light and safe automotive structures requires engineers to know how AHSS steels behaves under impact loading conditions. Automobile collisions are complex and characterized by highly nonlinear deformation processes. When the vehicle collides at a velocity over 60 km/h, the maximum strain rate of the material exceeds 1000 s^−1^ [19]. It leads to an increase in the plastic flow stress (strain rate hardening) and changes in the energy absorption characteristics of the main automobile body materials. In turn, an increase in the loading rate, from static to strongly dynamic, during the crushing of energy-absorbing structures could change the mechanism of deformation to a symmetric/mixed mode [20,21,22].

Many researchers have focused on investigations of the axial crushing of energy absorbers (EA), also called energy-absorbing thin-walled structures (TWS), made of high-strength steels, such as TRIP steel [23,24], DP steel [10,24], HSLA steel [10], and hot forming boron steel [25]. From these materials, various types of TWS were tested experimentally [5,10,26,27,28,29]. The most common cross-sections of TWS are top-hat [23,25,30], double-hat [31], circular [10], and square tubes [32]. As it is reported in [32,33], the square and hat-shaped energy absorbers made of AHS steels present excellent energy dissipation capability and reveal progressive folding. In turn, in works [33,34], TW sections manufactured from AHSS steel are compared with sections of common materials, and no significant material effect on the deformation mode is noticed.

The use of AHS steels brings many technological problems, including an increase in die wear, increased springback, weldability problems, flange stretching, edge cracking, fatigue strength, etc. [11,35]. A thin sheet joining technology is crucial to obtain the appropriate mechanical properties of joints and to decrease the process costs. The most widely used joining technology in automotive industry is resistance spot welding; however, it is increasingly supplanted by a rapidly developed laser welding technology. This method plays an important role in reducing the car body weight, reducing costs, and improving the crashworthiness, strength, and rigidity of the body-in-white structure [36,37]. Many works are based on TWS made with resistance spot welding; however, only a few papers discuss the behavior of the laser welding joint between profiles of AHSS thin-walled section under axial crushing load [32,38].

Docol 1000DP and Docol 1200M are commercial, cold-rolled steels widely used in new design in the automotive industry, in safety and saving parts of body-in-white applications. Docol 1000DP belongs to the group of dual-phase steel with a microstructure which consists of a soft ferritic matrix containing islands of martensite. Docol 1200M is martensitic steel with mainly martensitic microstructure with a small portion of ferritic and bainite [18]. Many authors have shown that dual-phase steel presents favorable crashworthiness properties in a wide range of strain rates [39,40,41,42]; however, the high strain rate response of martensitic steel has not been sufficiently described in the literature. In work [19], the authors present the results of a dynamic test of 1200M steel in the of strain rate range of 10^−3^–10^3^ s^−1^, which showed the negative strain rate sensitivity of the described steel grade. It is in contrast to other works [43,44,45], which reveal the positive strain rate sensitivity of the commercial martensitic steel.

Given the small number of works on the mechanical behavior of laser-welded joints in the Docol 1000DP and Docol 1200M steels sheets under impact loadings, it is important to consider the crushing behavior of the square tube absorbers fabricated from the above-mentioned AHS steels during axial crushing loading. Therefore, the aim of the present study is a crash response analysis of laser-welded square tube absorbers made of sheets of the Docol 1000DP and Docol 1200M steels at quasi-static and impact loading conditions. It is emphasized that a new approach has been proposed to study the tube absorber under impact loading using a direct impact Hopkinson (DIH) technique, which allows achieving a higher impact velocity (above 20 m/s) than a drop weight tower device.

The paper is organized as follows: Section 2 is devoted to a description of the above-mentioned AHS steels including their chemical composition and microstructure features. The details related to testing procedures of the mechanical response of the tested steels, the manufacturing process of small-scale tube absorbers with a square cross-section, and experimental procedures of crashing tests under the quasi-static loading as well as the dynamic loading with the use of a direct impact Hopkinson pressure bar technique are also presented in Section 2. Section 3 presents the obtained results on mechanical properties of the Docol 1000DP and Docol 1200M steels determined under a quasi-static ($\dot{\varepsilon }$ = 10^−3^ and 10^0^ s^−1^) and a high strain rate ($\dot{\varepsilon }$ > 10^3^ s^−1^) loading condition and also describes microstructure of laser-welded joints and microhardness profiles. The crash response of the square tube absorbers and analysis of the resulting cracks are also presented in Section 3.

## 2. Materials and Experimental Methods

### 2.1. Materials

The materials studied in this work are two grades of commercial high-strength steels: Docol 1000DP and Docol 1200M. Cold-rolled and non-galvanized 2 mm-thick steel sheets as received with the chemical compositions listed in Table 1 were used for further processing.

For the microscopic analysis purpose, the as-received material specimens and weld cross-sections samples were prepared according to the standard procedure and etched with 2% nital solution. The microstructure examination was carried out using a light microscope (VHX6000, Keyence, Mechelen, Belgium). The observations of as-received specimens revealed the presence of martensite and ferrite in the 1000DP steel (Figure 1a). The dominant in the material martensite forms a contiguous microstructure and encompasses almost all ferrite grains. Unlike the martensite grains, ferrite grains vary in size. In turn, the 1200M steel consists of predominantly martensite phase with the small proportions of bainite and ferrite (Figure 1b). The microstructure of the material is more homogeneous in terms of grain size than the 1000DP steel. In the case of both steels, martensite ensures the material strength, while ferrite is responsible for ductile properties of materials, depending on their content in the materials. Based on the multiphase structure of both steel grades, both materials are classified to the AHSS group.

### 2.2. Mechanical Testing Procedure

To characterize the mechanical properties of the studied materials, hardness measurement and tensile strength tests were performed. Microhardness tests of cross-sections of laser-welded joints were performed at the load of 100 g with a holding time of 10 s using a Vickers hardness tester (Qness 10 CHD Master, QATM, Golling, Austria). In accordance with EN ISO 6507-1:2018 [47], all indents were spaced 80 μm apart to avoid any effect of deformation fields caused by adjacent indentations.

The strength testing of the materials as-received was carried out for three ranges of strain rates, i.e., 10^−3^, 10^0^, and above 10^3^ s^−1^. Two quasi-static strain rates were performed with the strength machine(Criterion C45, MTS System Corporation, Berlin, Germany), whereas high strain rate experiments were performed with the use of a tensile split Hopkinson pressure bar (SHPB) setup, which is similar to the conventional compressive SHPB arrangement, and its operation principle is based on the solution proposed in [48]. The basic parameters of the bar system shown in Figure 2 are as follows: loading and transmitted bar length—2000 mm and 1200 mm, respectively, striker bar length—450 mm, the diameter of all bars—12 mm. The bars were made of C45 steel (nominal quasi-static yield strength *R*_0.2_ = 710 MPa, sound speed *C_o_* = 5140 m/s). 

In both cases of strength tests, the same specimen geometry was applied (Figure 3a). This approach allows for a direct comparison of the ductility material parameters determined under different loading conditions. The shape of the specimen with a 7 mm gauge length results from SHPB requirements such as the limitation of inertial forces acting on the research system and achievement of a stress equilibrium state during the dynamic tension [49]. Five tensile strength specimens for each type of test (i.e., fifteen samples for the given steel grade) were made using the wire electrical discharge machining. All the specimens were prepared in such a manner that the tensile axis is perpendicular to the rolling direction.

A pair of special grips with gripping cavities was developed to ensure the proper mounting of specimens between SHPB bars (Figure 3b). Grips were screwed to the SHPB bars with the use of Teflon tape. It was found experimentally that the developed clamping system introduce small fluctuations (noise) in the signals recorded by the bars strain gages.

To minimize the wave dispersion by damping the Pochhammer–Chree high-frequency oscillations and to facilitate the stress equilibrium, a pulse-shaping technique was used [50]. The copper pulse shaper with a diameter of 3 mm and the 0.2 mm thicknesses guaranteed damping the high-frequency oscillations and achieving the dynamic stress equilibrium in the specimen under an averaged impact striker velocity of 13 m/s.

As a measure of ductility of the tested steels, there were applied two parameters, i.e., $\varepsilon_{f}$—fracture strain, and $\varepsilon_{u}$—uniform strain, which were calculated on the basis of post-test measurements of the gauge length and cross-sectional dimensions of recovered parts of the tensile specimen. Parameter $\varepsilon_{f}$, called also total elongation, was calculated from a widely known relation, whereas $\varepsilon_{u}$ was determined using the following formula: $\varepsilon_{u}=\left(A_{0}-A\right)/A$, where $A_{0}$ and A are the initial and the deformed cross-sectional areas, respectively. It should be noted that the cross-sectional area A was determined in the middle of the fractured gauge length of the tensile specimen—that is, between the neck and the gauge length end of specimen. Measurements with the ±5 μm accuracy were conducted on all pieces of specimens with the use of an optical comparator.

### 2.3. Square Tube Absorber Fabrication

A step-by-step manufacturing process of miniature absorbers is shown in Figure 4. First, semi-finished sheet metal blanks were laser cut to rectangular dimensions of 120 × 42 mm (Figure 4a). Due to limitations associated with the maximum initial kinetic energy of the direct impact Hopkinson stand to crush the column absorbers, the sheets were ground from 2.0 to 0.6 mm of the thickness with a cooling liquid from both sides to prevent overheating. Next, sheet cuttings were bent on a servo-electric press brake with a bending tool radius of 4 mm to obtain a U-shaped profile (Figure 4b). Subsequently, two U-shaped pieces of the square tube absorber were joined by means of spot welds made with MAG (Metal Active Gas) technology to maintain precision during laser welding.

It should be emphasized that it is crucial to maintain a gap between the welded components with a constant width and not more than 5% of the thickness of the welded material [51]. Therefore, it was a strenuous task to ensure that the gap between the edges of U-shaped profiles was not greater than 0.05 mm after the initial weld operation. However, due to manufacturing errors of all the preceding production processes before laser welding, it was decided to use the wobbling system in the circle mode, which allows for an increase in the gap width criterion from 0.03 to 0.06 mm. The laser-welding process was carried out using an IPG multi-axis welding workstation (Figure 4c). Welding parameters guaranteeing the high strength of welding were selected after numerous trials and are listed in Table 2. It should be noted that weld lines were performed perpendicularly to the rolling direction of the sheets, which means that the most unfavorable configuration was assumed in view of the toughness of the weld joint.

After the laser welding process, the initial triggers were formed with a servo-electric press brake with special developed tools. Due to the different springback parameters of the investigated material, the repeatability of the trigger geometry was within the range of +/− 0.5 mm. The final stages of production included cutting off the square tube ends with WEDM technology to remove spot welds and to ensure parallelism of the absorber front surfaces. For each steel grade, two square tubes were used for both the quasi-static and the dynamic crush experiments.

### 2.4. Crash Tests Procedure

All the square tube absorbers were tested for quasi-static and dynamic axial crushing at the room temperature. The quasi-static crush tests were carried out with the use of an electromechanical testing machine (Criterion C45, MTS System Corporation, Berlin, Germany) (Figure 5) that was also used in the mechanical characterization of the tested AHS steels. A constant base traverse velocity of 0.5 mm/s was utilized, and the tests were stopped after reaching a crushing distance of 75 mm (approximately 1/4 of the initial length of the energy absorber). Additionally, a digital camera was used to record the crushing process during the tests. The videos were post-processed to locate the initiation of the failure crack within the energy absorber and to identify the crushing mechanism.

Dynamic crushing experiments were performed using a modified split Hopkinson pressure bar setup, often called a direct impact Hopkinson (DIH) configuration. The loading condition of the DIH configuration is similar to the testing condition of drop tower system; however, the DIH system enables a higher crushing velocity above 20 m/s. Moreover, the DIH technique has been widely used in studies concerned the strain rate effects of metallic foams [52], regular cellular structures [53], and other mechanical absorbing systems [54]. The main difference between the DIH configurations used in the present work and the standard SHPB configuration is the position of the tested specimen. It is placed on the front surface of the input bar, in front of a muzzle barrel of the gas gun system, which launches a striker bar (Figure 6).

In the present work, a large-scale SHPB system, i.e., bar diameters of 40 mm, input and output bar lengths of 3 m each, and 360 mm long striker bar with diameter 36 mm (approximately 2.86 kg), was used. All the bars were made of C40 steel, which was characterized by a sound speed of 5140 m/s, Young’s modulus of 207.5 GPa, and the yield strength of 460 MPa.

The strain ε(t) on the input bar used to determine a crushing force F(t) was measured using a full-bridge strain-gage circuit, in which strain gauges were glued 1500 mm away from the front input bar end. The crushing force F(t) was calculated using the following relation Equation (1):(1)$$F\left(t\right)=EA_{0}\varepsilon \left(t\right),$$
where E and $A_{0}$ denote the Young modulus and the cross-section of the input bar, respectively.

In turn, shortening (deformation) of the tube absorber during the dynamic crushing was calculated based on the direct measurement of the striker bar displacement d(t). The displacement was captured by a high-speed camera (v1612, PHANTOM, Leicester, UK) at 95,000 fps and at a resolution of 512 × 256 dpi. To achieve a high measuring accuracy, a characteristic fiducial marker symbol (Secchi disk; Figure 5) was glued on the lateral side of the striker, and Tema Motion software(Tema Classic 2D, Image System Motion Analysis, Linkoping, Sweden) was used to process the raw record videos and to determine the displacement-time curves d(t). The striker bar velocity prior to the impact was also determined using the above-mentioned camera and software. The striker impact velocity varied between 25 and 28 m/s. To protect the contact surfaces of the striker and the input bars against damage, 2 mm thick and 35 mm diameter steel disks were glued to both ends of the tube absorber with the use of cyanoacrylate adhesive (Figure 4). The adhesive was also applied to attach the tube absorber samples to the front end of the input bar.

## 3. Results and Discussion

### 3.1. Microstructure and Microhardness

To assess the quality of the laser-welded joints in tube absorbers, the visual inspection in accordance with EN ISO 13919-1:2020 [55] was conducted at first. It revealed no welding defects, such as cracks, porosity, or a lack of joint penetration, coming out to the surface. Next, the macrostructural examination of laser-welded joints was performed. The overall views of the weld cross-section of the Docol 1000DP and 1200M are shown in Figure 7a,b, respectively. It is observed that the width of the fusion zone (FZ) in both material grades joints is similar and results mainly from use of a laser-welding wobbling system. The heat-affected zone (HAZ) is close to 0.3 mm on both sides of the welded joint. The microstructure of the fusion zone for both steels consists mainly of a hard martensitic structure; however, the grain size is bigger in the Docol 1200M than in 1000DP steel.

The microhardness profiles on the weld cross-section presented in Figure 8 show a significant difference between hardness of the Docol 1000DP steel (approximately 330 HV0.1) and hardness of the fusion zone (approximately 420 HV0.1), whereas the hardness of Docol 1200M at the fusion zone and of the base material is almost the same and equals approximately 400 HV0.1 (Figure 8). These observations provide that rapid weld cooling during the laser welding of Docol 1000DP leads to increasing the martensite volume fraction in the FZ region, which is typical for dual-phase steels. In both cases of the AHS steels, the heat-affected zones hardness is reduced in comparison to the base material, i.e., the microhardness drop is 60 and 115 HV0.1 for Docol 1000DP and 1200M, respectively. The higher drop in hardness in HAZ of Docol 1200M results from the tempering process, which forms a softened zone. This zone may affect the increase in plastic properties of the joints.

### 3.2. Quasi-Static and High Strain Rate Behavior

The engineering stress–strain curves for three ranges of strain rates are shown in Figure 9a,b for Docol 1000DP and Docol 1200M, respectively. The presented stress–strain curves are representative (averaged) curves drawn based on a five experimental datasets. It should be noted that high strain rate curves exhibited oscillations, which were not a real mechanical response of the materials, but they resulted from the technical limitations of the SHPB technique. Therefore, high strain rate curves presented in Figure 9 were smoothed using the SciDAVis software using the Savitzky–Golay method.

For the quasi-static loading conditions, the tested steels exhibit a slight difference in mechanical response depending on the applied strain rates. The stress–strain curves for a strain rate of 10^0^ s^−1^ show a small increase in the plastic flow stress comparing to the curves obtained at 10^−3^ s^−1^; i.e., stress peaks corresponding to the ultimate tensile strength (UTS) are equal to 1038 and 1132 MPa for Docol 1000DP, and they are 1241 and 1267 MPa for Docol 1200M. These values indicate that Docol 1000DP reveals a higher strain rate sensitivity than Docol 1200M in the considered range of strain rate. Compared to the quasi-static tension, the strength of steels tested under high strain rates of 1300 and 1050 s^−1^ is larger. The stress peaks for Docol 1000DP and 1200M achieve levels of 1303 and 1452 MPa, respectively. In the cases of both steels, an increase in the plastic flow stress is almost the same compared to the quasi-static tension and it is equal to approximately 17%. It shows relatively high positive strain rate sensitivity of the tested steels, and it is in contrast to the results presented in work [19], in which the negative strain rate sensitivity of the 1200M steel is demonstrated.

The ductile behavior of the considered AHS steels is interesting from a cognitive point of view. A decrease in the elongation is usually observed at higher strain rates, which is indicative of material susceptibility to brittle fracture. On the contrary, the tested steels show relatively high ductility under dynamic loadings. As shown in Table 3, ductility parameters, i.e., $\varepsilon_{f}$—fracture strain, and $\varepsilon_{u}$—uniform strain, decrease initially or remain at the same level at strain rate of 10^0^ s^−1^, and next, they increase slightly at a strain rate above 10^3^ s^−1^.

From a point of view of crashworthiness, the above-mentioned features observed in the tested steels, expressed by their satisfactory toughness under the high strain rate loading, are highly desirable. It should be assumed that absorber structures made of these steels will also have a high capability of mechanical energy absorption.

As an additional comment, we require also the values of $\varepsilon_{f}$ and $\varepsilon_{u}$ obtained from the quasi-static and the high strain rate experiments. In the literature [56,57], it is reported that the elongation *A*_80_ for Docol 1000DP and Docol 1200M steels, determined in accordance with quasi-static tensile test standards, is 0.08 and 0.05, respectively. Compared to the values of $\varepsilon_{f}$ collected in Table 3, significant differences can be found, because a total elongation, corresponding to $\varepsilon_{f}$, is much higher and amounts to 0.21 and 0.23, respectively. These differences result from the geometry of tensile specimens applied at the strength testing. In the present work, the tensile specimens with a short 7 mm gauge length were used, whereas the data reported in the literature were obtained for the flat specimens with a long 80 mm gauge length. Moreover, in the case of short gauge length specimens, the contribution of necking strain to total specimen elongation is significant, especially for materials demonstrating low strain hardening. The studied steels belong to this material group, and they manifest high strain localization in the form of necking. Therefore, uniform strain $\varepsilon_{u}$ seems to be a more suitable parameter for evaluation of the ductile behavior of materials.

It should be emphasized that the use of the short-length specimens may also cause some distortion of information about the strain-hardening behavior. From the stress–strain curves presented in Figure 9, it may be observed that Docol 1000DP and Docol 1200M exhibit strain softening, thermal softening, or strain-rate softening effects. Literature reports do not confirm these observations for the strain rate range applied in the present work. As above-mentioned, a significant contribution of necking to the deformation process of a short-length specimen causes a greater part of the curve to correspond to the deformation stage at which the strain localization developed and a non-uniform stress state occurred. Hance, the assessment of the hardening behavior of the tested materials on the basis of the results obtained from the short-length specimen may lead to incorrect conclusions.

### 3.3. Quasi-Static Crush Response

The deformation processes of the square tube absorbers under the quasi-static axial crushing are presented in Figure 10. When the upper traverse grip of the strength machine contacts the absorber, the first pair of folds begins to form. The crush triggers were oriented to the inside of the non-welded walls, thus enforcing the folding inwards, and the folds formed on laser-welded walls bent to the outside. The maximum force initiating the progressive folding of the square tube absorber is strongly dependent on the yield point of the material; thus, the registered peak force is 35% higher for 1200M steel, for which the yield point is 25% higher than for 1000DP steel. It was found that the glue connection between the tube absorber and the protective disc can also affect the post-peak slope of the crush force–displacement curve, and its premature failure allows for small relative movements, decreasing the axial stiffness. To limit the influence of the boundary effects, both the upper and the lower discs should be permanently connected to the tube absorber with welding or shaped connections, which ensure that the measured response results purely from the miniature absorber geometry [30].

The local peaks of force corresponding to the moments at which the formed folds began to lose stability do not exceed 15 and 20 kN for the 1000DP and 1200M steels, respectively. Each peak corresponds to one pair of the formed folds. In this phase of compression, a relatively constant force plateau is observed. After the experimental tests, a slight springback of 3 mm for the Docol 1000DP and 4 mm for the Docol 1200M absorbers was observed.

Depending on the folding direction, inside or outside of tube absorber cross-section, the cracks propagated in the middle of the fillet (inside the folding) or on the transition of the fillet into the flat surface (outside the folding). In this area, the local bending radius is the lowest, which results in intense stretching of the sheet surface. The number of actual cracks observed in the tube absorber was lower for the 1200M steel; however, they were characterized by a lower length and different orientation. From observations of the longitudinal cross-section of the crushed absorbers (Figure 11), it was found that folds of the Docol 1200M tube absorber have a lower depth in comparison to Docol 1000DP (*m* > *n*), which means that the stiffness and springback of the Docol 1200M tube absorber are higher than the Docol 1000DP one. In the quasi-static tests, no separation of welding was noticed for either of the absorber materials. Only single and insignificant cracks formed after the full crushing of the tube absorbers was observed.

### 3.4. Dynamic Crush Response

The process of fold formation, during the DIH test, in the absorbers is similar to the quasi-static fold formation. The stages of the impact crushing process are shown in Figure 12. The main difference between static and impact deformation is related to the width of the second fold, which in impact tests is significantly narrower. This is a result of inertial effects delaying the formation of the third couple of folds. As opposed to static compression, when the striker contacts the absorber, the glued connections between the absorber and the washer discs fail first at both ends of the tube absorber. Then, the first pair of folds on the welded walls is formed on the outside. At the same time, a pair of folds is formed on the inside of the non-welded walls. Due to the ground surface of the absorber walls, the vibration wave of the not-yet-crushed surfaces of the absorber walls is visible during the entire crushing process.

In Figure 12a,b, a crush force calculated on the base of strain measurements with the use of the strain gauges placed on the input bar is presented as a function of the crushing distance for both tube absorber materials. Except for the initial force peak, during the progressive deformation of the tube absorber, the additional outstanding peaks (for displacement of 13, 26, 39, and 52 mm—Docol 1000DP) were observed. They result from the material folding but also from the reflection of the initial stress wave from the end of the input bar. Hence, negative values of forces observed in the integrated crush force–displacement curves. This phenomenon is a main disadvantage of the proposed direct impact Hopkinson method, which is related to the geometry of the stand. To prevent this reflection, the length of the initial bar should be increased 12 times.

Figure 13 shows the areas of cracks occurring in the crushed absorbers. Cracks occurred mainly in the corners, in place of stress accumulation, similar to cracks formed under the quasi-static conditions. In impact conditions, a smaller number of cracks was also observed, which can be correlated with an increase in the tested steels strength and ductility along with an increase in a strain rate [41]. In the case of impact compression, longitudinal cracks in the weld on the second fold were observed during the formation of the third pairs of folds. This is the effect of a folding delay due to inertia. Therefore, the second fold deforms much more, and the bending corner radius decreases, intensifying the stretching of the welded butt-joint in the normal direction. Deformation of the welded joint is smaller than the maximum deformation at the edges of the folds; however, because a weld tends to reduce the ductility of the material, the amount of deformation is sufficient to create cracks.

### 3.5. Mechnical Energy Absorption Capability

One of the main parameters characterizing the column energy absorber is the absorption energy ($E_{a}$). The ability of the structure to absorb energy during the impact is calculated using Equation (2) [25], where F(x) is the crush force and w is the displacement up to which the energy absorption.
(2)$$E_{a}\left(x\right)=\int_{0}^{w}F\left(x\right)dx,$$

Energy absorption–displacement curves shown in Figure 14a,b were determined on the basis of the modified crushing force curves, which were smooth by removal of wave disturbances generated during the tube absorber crushing.

Under quasi-static conditions, the value of energy absorption by absorbers made of Docol 1200M steel is approximately 21% higher than that of absorbers made of Docol 1000DP steel; under the impact conditions, this ratio is 9%. The difference between crushing speeds for Docol 1200M steel in the energy value is approximately 11%, while for Docol 1000DP steel, it is 23%, which confirms the strain rate sensitivity of the dual-phase steels.

A graphic representation of the mean crushing force ($F_{mean})$ for both steel grades is given in Figure 14c,d. $F_{mean}$ was calculated with Equation (3). The difference between $F_{mean}$ levels is less visible in the quasi-static conditions than in the impact conditions. The values of the mean crushing force are also presented in Table 4.
(3)$$F_{mean}=\frac{E\left(w\right)}{w}$$

The crushing force efficiency (CFE) factor is a measure of the stability of the energy-absorption process and was calculated with Equation (4), where $F_{mean}$ is the mean crushing force and $F_{peak}$ is the maximum crushing force. Good energy absorbers should vary less between the mean crushing force and the peak force [58].
(4)$$CFE=\frac{F_{mean}}{F_{peak}}$$

For both steel grades, the CFE was calculated and is shown in Table 4. Under quasi-static loading conditions, the CFE factor is more favorable for absorbers made of Docol 1000DP steel, while under the impact loading conditions, the factor is comparable for both steel grades. For the same absorber geometry, as the material strength increases, the value of CFE increases, and a similar relationship was observed in work [25].

The specific energy absorbed (SEA) was calculated for both steel grades with Equation (5), where m is a mass unit of energy absorber. The SEA coefficient allows comparing energy absorbers with different weights; the greater the SEA is, the greater the crashworthiness capacity of the structure [25].
(5)$$SEA=\frac{E_{a}}{m},$$

The specific energy absorbed is approximately 8% higher in quasi-static conditions and 9% in dynamic conditions for Docol 1200M compared with Docol 1000DP. The SEA level may be comparable to the results found in other research [38]. The crushing performance is more efficient for the energy absorbers made of Docol 1000DP steel. Considering the absorber production stage, Docol 1000DP steel is more often used for the production of column energy absorbers. The low ductility of Docol 1200M steel causes many problems associated with the stamping of mechanical energy-absorbing components. Therefore, many components made of this steel, for example, elements such as bumper reinforcements or side impact door beams, are manufactured in a simple shape using a roll-forming technology.

## 4. Conclusions

The paper presents the results of the experimental studies of crushing the laser-welded square tube absorbers made of two high-strength steel grades: Docol 1000DP and Docol 1200M. Based on the obtained results, the following conclusions can be drawn:Docol 1000DP steel demonstrates a higher strain rate sensitivity than that of Docol 1200M steel. The strain rate loading condition increases the ductility of both steel grades.A well-made laser-welded butt joint does not adversely affect the crushing process of the absorber; in most absorbers, no cracks in the welded joints were noticed.Laser welding technology enables the production of high-strength welded joints, which are crucial in the crash performance of column energy absorbers.The use of a more durable method to connect the tube absorber and the protective discs would increase the stability of the absorber crushing process. Welding should be more reliable than adhesive glue connection.The influence of the strain rate and inertia forces on the energy absorption and the crushing process were observed. The difference is mainly due to the material strain rate sensitivity.Under quasi-static and impact loading conditions, SEA is greater for Docol 1200M in comparison in Docol 1200M, whereas CFE is more favorable for Docol 1000DP. In engineering, column energy absorbers are more preferred with greater CFE; for that reason, Docol 1000DP is the more preferred material used in design axial crash energy absorbers.The use of the DIH method may be an appropriate approach for investigation of the crushing behavior of model tube absorbers.

## Figures and Tables

**Figure 1 materials-14-02808-f001:**
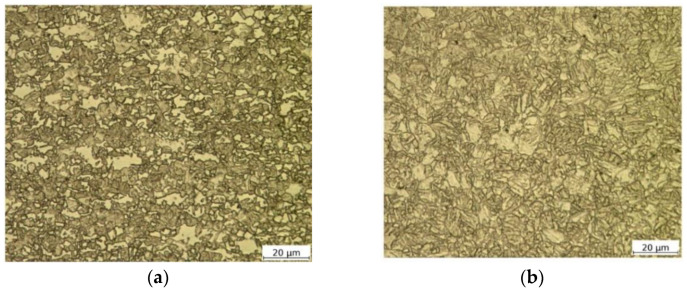
Optical microscopy images demonstrating the microstructures of the as-received materials: Docol 1000DP (**a**) and Docol 1200M (**b**).

**Figure 2 materials-14-02808-f002:**
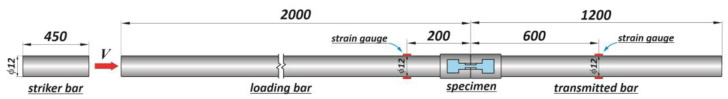
A scheme of the bar system of the tensile SHPB arrangement.

**Figure 3 materials-14-02808-f003:**
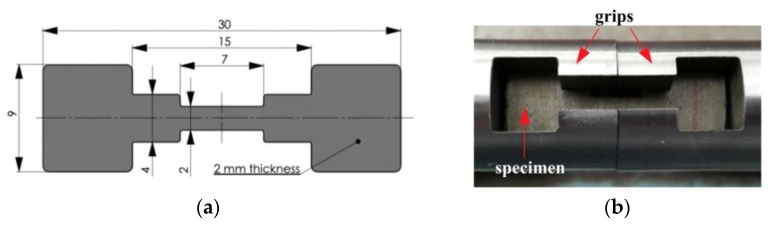
Geometry and dimension of the tensile strength specimen used in the quasi-static and high strain rate tests (**a**); a view of the specimen mounted in the grips of the SHPB system (**b**).

**Figure 4 materials-14-02808-f004:**
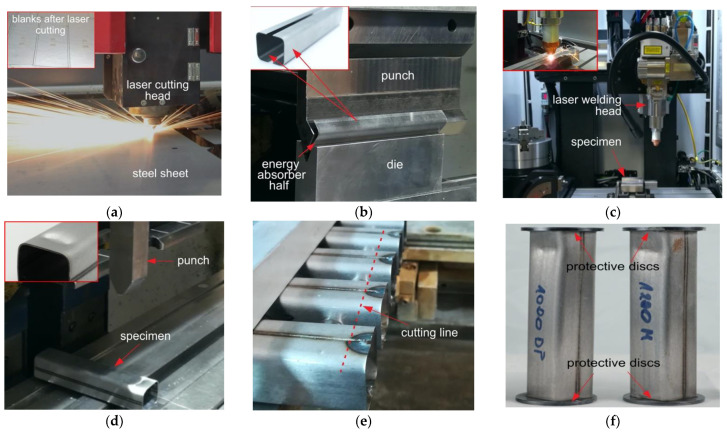
A scheme of the manufacturing process of energy absorbers: (**a**) laser sheet cutting; (**b**) press brake bending of geometries of the halves; (**c**) fiber laser welding of the square tube absorber; (**d**) formation of the initial triggers; (**e**) WEDM cutting off the square tube ends; (**f**) overview of the final square tube absorbers with the protective discs.

**Figure 5 materials-14-02808-f005:**
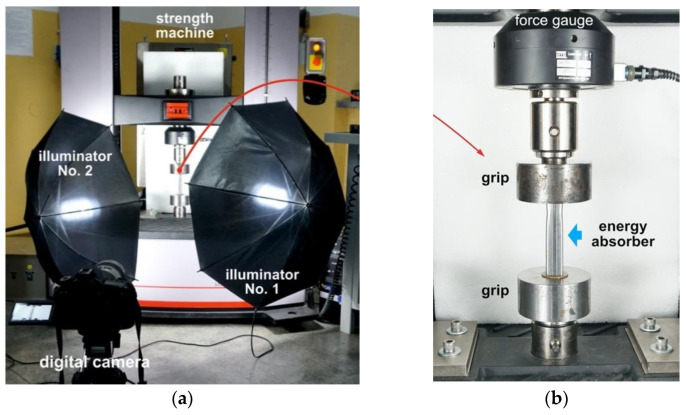
A view of the experimental setup for quasi-static crushing testing of square tube absorber: (**a**) overall view of the electromechanical strength machine and the image recording system (digital camera and two illuminators); (**b**) energy absorber positioned on the electromechanical strength machine.

**Figure 6 materials-14-02808-f006:**
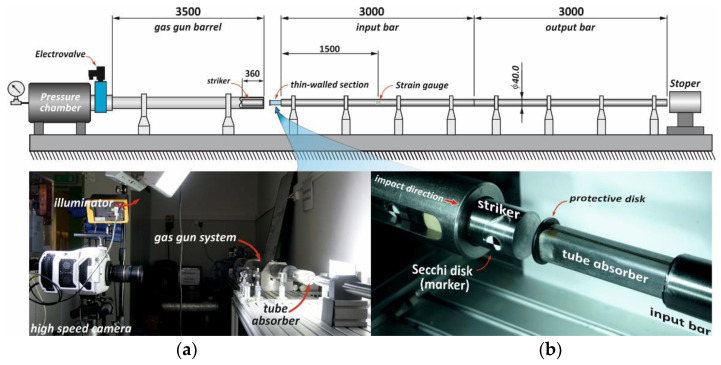
A scheme of the direct impact Hopkinson setup used to study the square tube absorbers and a view of selected setup components invisible in the scheme: (**a**) high-speed image recording system (Phantom v1612 camera, LED illuminators); (**b**) tube absorber mounted on the front surface of the input bar.

**Figure 7 materials-14-02808-f007:**
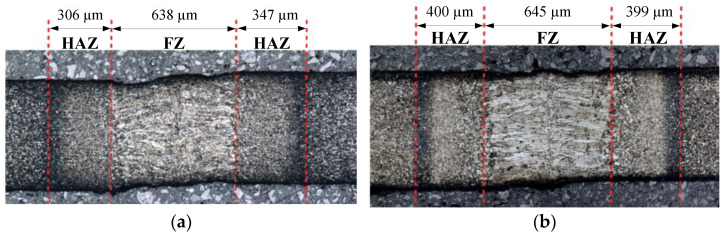
Cross-section macrostructures of laser-welded joints and main dimensions of the fusion zone (FZ) and the heat-affected zone (HAZ): (**a**) Docol 1000DP; (**b**) Docol 1200M.

**Figure 8 materials-14-02808-f008:**
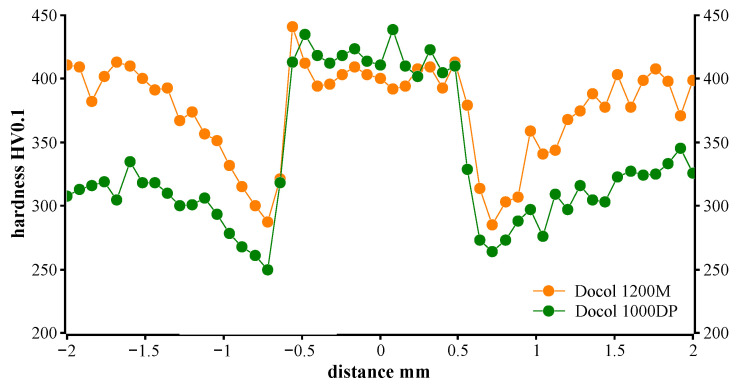
A microhardness profile across laser-welded joints of Docol 1000DP and Docol 1200M.

**Figure 9 materials-14-02808-f009:**
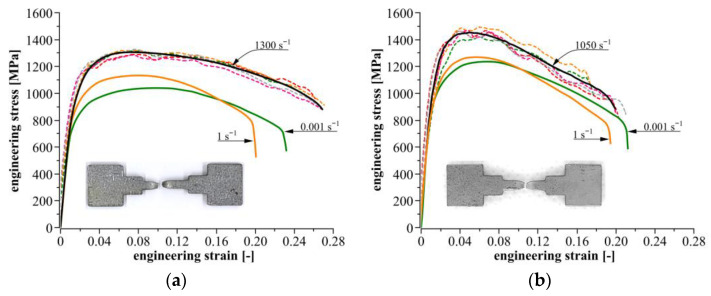
Quasi-static and high strain rate stress–strain curves determined from tensile strength experiments for: (**a**) Docol 1000DP; (**b**) Docol 1200M.

**Figure 10 materials-14-02808-f010:**
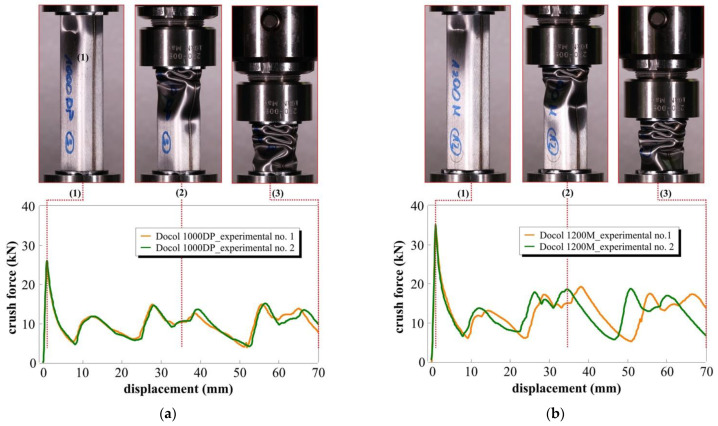
Square tube absorber deformation (top) obtained from quasi-static axial crush experiments and crush force–displacement curves (down) for (**a**) Docol 1000DP; (**b**) Docol 1200M materials (red dotted lines denote selected phases of crushing, i.e., (**1**) peak force, (**2**) in the middle of the crushing period, and (**3**) the end of the crushing process).

**Figure 11 materials-14-02808-f011:**
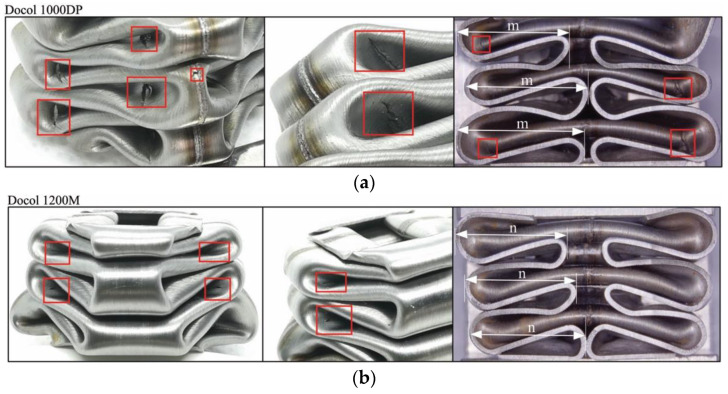
A view of crushed square tubes and the resulting cracks after quasi-static tests with marked folding depth: (**a**) Docol 1000DP; (**b**) Docol 1200M (cracks are marked with red rectangles).

**Figure 12 materials-14-02808-f012:**
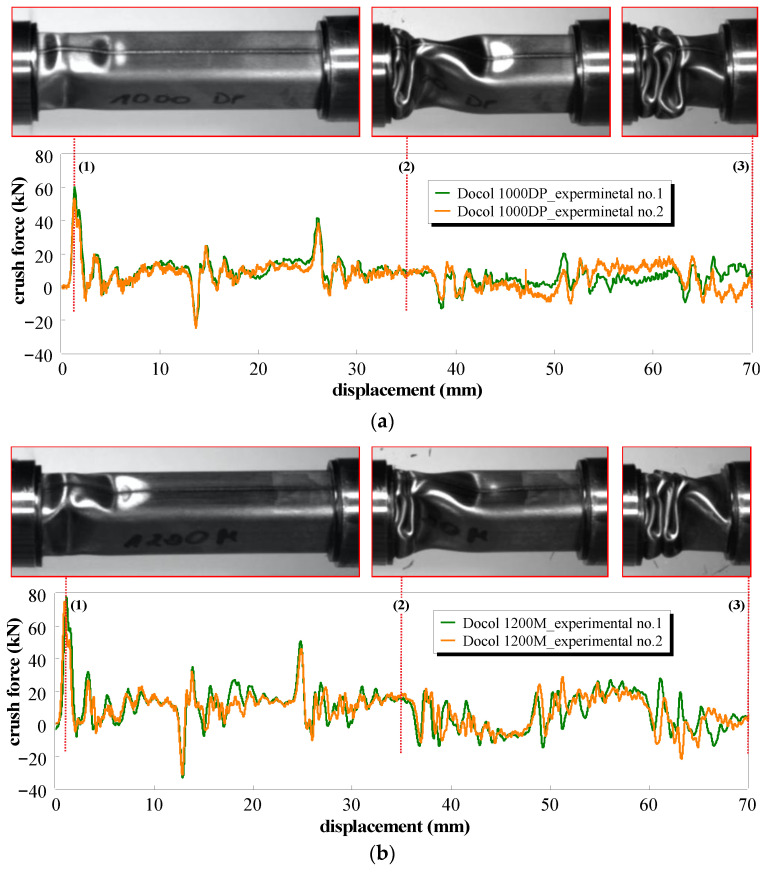
Square tube absorber deformation (top) obtained from dynamic axial crush experiments and crush force–displacement curves (down) for (**a**) Docol 1000DP; (**b**) Docol 1200M materials (red dotted lines denote selected phases of dynamic crushing, i.e., (**1**) peak force, (**2**) in the middle of the crushing period, and (**3**) the end of crushing process).

**Figure 13 materials-14-02808-f013:**
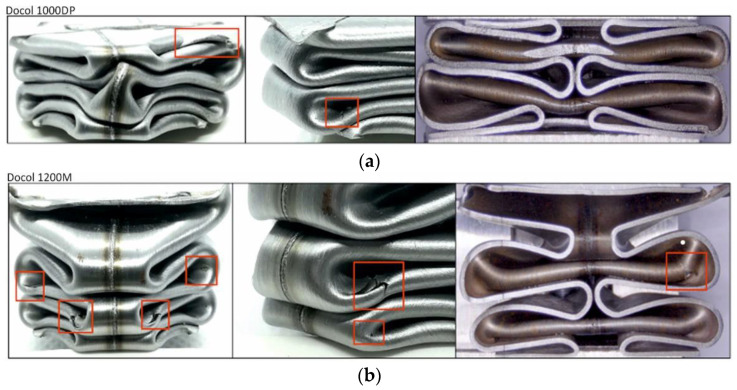
A view of the dynamically crushed square tubes and the resulting cracks: (**a**) Docol 1000DP; (**b**) Docol 1200M (cracks are marked with red rectangles).

**Figure 14 materials-14-02808-f014:**
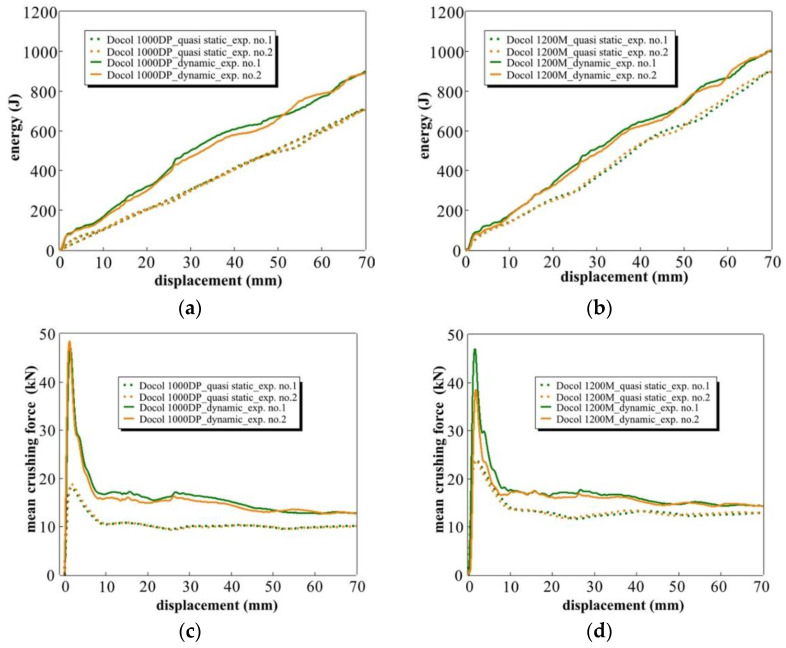
The comparison of the absorbed energy (top) and the mean crushing force (down) calculated from quasi-static and dynamic crushing experiments for: (**a**,**c**) Docol 1000DP; (**b**,**d**) Docol 1200M.

**Table 1 materials-14-02808-t001:** Chemical compositions of Docol 1000DP and Docol 1200M (in wt %) [46].

Material	C	Si	Mn	P	Cr	Ni	Al	Cu	Nb	V
Docol 1000DP	0.130	0.21	1.47	0.011	0.02	0.03	0.047	0.01	0.017	0.01
Docol 1200M	0.111	0.21	1.59	0.009	0.02	0.04	0.04	0.01	0.000	0.02

**Table 2 materials-14-02808-t002:** Laser welding parameters.

Parameter		Value
Beam power	(W)	450
Fiber diameter	(µm)	50
Spot diameter	(µm)	100
Wobble mode		Circle
Wobble frequency	(Hz)	250
Wobble diameter	(mm)	0.5
Focal length (mm)		200
Welding speed	(mm/s)	37.5
Joint type		Butt
Sheet thickness	(mm)	0.6

**Table 3 materials-14-02808-t003:** Ductility parameters for Docol 1000DP and Docol 1200M steels.

**Material**	$\dot{\varepsilon }=10^{-3}\;s^{-1}$		$\dot{\varepsilon }=10^{0}\;s^{-1}$		$\dot{\varepsilon }=10^{3}\;s^{-1}$	
	$\varepsilon_{f}*$	$\varepsilon_{u}*$	$\varepsilon_{f}$	$\varepsilon_{u}$	$\varepsilon_{f}$	$\varepsilon_{u}$
Docol 1000DP	0.23	0.16	0.19	0.11	0.26	0.18
Docol 1200M	0.21	0.09	0.19	0.09	0.20	0.15

* $\varepsilon_{f}$—fracture strain; $\varepsilon_{u}$—uniform strain.

**Table 4 materials-14-02808-t004:** Comparison of the energy-absorbing parameters of Docol 1000DP and Docol 1200M.

Material	Absorbed Energy at 70 mm [J]	Peak Force [kN]	Mean Force [kN]	CFE	SEA [kJ/kg]
	Quasi-static conditions				
1000DP_Exp. No. 1	710	26.1	10.2	0.40	18.9
1000DP_Exp. No. 2	707	25.3	10.1	0.40	18.8
1200M_Exp. No. 1	902	35.1	13.5	0.38	24.0
1200M_Exp. No. 2	897	34.2	12.9	0.38	23.9
	Dynamic conditions				
1000DP_Exp. No. 1	905	60.4	13.1	0.22	24.0
1000DP_Exp. No. 2	935	53.6	13.8	0.26	24.9
1200M_Exp. No. 1	1007	79.8	14.3	0.18	26.8
1200M_Exp. No. 2	1009	77.6	14.3	0.18	26.8

## Data Availability

The data presented in this study are available on request from the corresponding author.

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
