# Peer review of "Crash Response of Laser-Welded Energy Absorbers Made of Docol 1000DP and Docol 1200M Steels"

_materials, 2021, doi:10.3390/ma14112808_

Round 1
Reviewer 1 Report
The authors examined the crush response with various strain rates, which is an interesting work. There have few points I would like to further understand.
- Regarding hardness distribution, did the authors check the microstructure near the HAZ in detail? The HAZ was possibly subjected to recrystallization and high-temp tempering, which is intrigued to know what kind of phase comprised in it after the welding.
- Please explain why engineering strain response is seemingly regardless of strain rate in these two different kinds of steels (Fig. 9). For example, why quasi-static rate (0.001/s) is with an intermediate engineering strain between mid- (1/s) and high (1000/s). Did it has something to do with different microstructure distributions?
Author Response
Dear Reviewer,
Thank you for your comments concerning our manuscript entitled “ Crash Response of Laser-Welded Energy Absorbers Made of Docol 1000DP and Docol 1200M Steels ” (Manuscript ID: materials-1209412). Those comments are all valuable and very helpful for revising and improving our paper.
We have made a correction which we hope meet with approval. Revised portions are highlighted in yellow in the paper.
Answers
- Regarding hardness distribution, did the authors check the microstructure near the HAZ in detail? The HAZ was possibly subjected to recrystallization and high-temp tempering, which is intrigued to know what kind of phase comprised in it after the welding.
Answer: The microstructure in the HAZ zone has not been thoroughly checked, however, taking into account the decrease in hardness in this zone, the material has tempered. Macroscopic examinations of laser welded joints will be developed in future research works.
- Please explain why engineering strain response is seemingly regardless of strain rate in these two different kinds of steels (Fig. 9). For example, why quasi-static rate (0.001/s) is with an intermediate engineering strain between mid- (1/s) and high (1000/s). Did it has something to do with different microstructure distributions?
Answer: Information on the mechanisms of deformation of AHSS at high strain rates is not well understood. Generally, it is believed that increase in ductility at high strain rate or strain rate independence is resulted in delay of strain-induced martensitic transformation of retained austenite. In other words, high strain rate conditions may delay of austenite-to-martensite transformation to greater strains, thereby extending the onset of necking, and consequently increases the ductility (total elongation). For quasi-static strain rate regime, austenite-to-martensite transformation occurs earlier, i.e. at lower strains, therefore decrease in ductility is observed due to higher volume fraction of formed martensite. The ductility increase at high-strain-rate loading condition is usually reported both dual-phase and martensitic steels.
After carefully studying the Reviewer comments, we have tried our best to revise and improve the manuscript and hope that the correction of manuscript will meet with approval.
Once again, thank you very much for your comments and suggestions.
Yours sincerely
Paweł Prochenka
Reviewer 2 Report
The current study investigates the crash response of steel absorbers that are laser welded. For this the authors carry out crash/impact tests using quasi static loading and dynamic impact loading at various speeds. The authors validate the laser welded steel specimens according to their absorbed energy, mean force and a crushing force efficiency factor. A
Abstract is written to good level but the authors are encouraged to consider the following to improve it: Please consider reviewing the abstract and highlight the novelty, major findings and conclusions.
Line 19 please consider removing this sentence “On the basis of the obtained results” as it does not add any value to the line
Line 25 “increase in ductility of both steel grades under the high strain rate loading conditions.” this sentence is a bit vague, so which of the two tested steels showed better ductility response?
Please avoid bulk citations unless you give all of them full credit somewhere else in the manuscript for example see lines 31, 36
At the end of the introduction before the layout of manuscript paragraph, the authors should attempt to answer the following questions: What is the research gap did you find from the previous researchers in your field? Mention it properly. It will improve the strength of the article.
For table 2 what is the reason behind choosing those specific laser welding parameters? Is it based on past literature or recommendation from industry or is it based on the machine specifications?
How many samples were made for each type of steel tested in this work?
Figure 5 it is advised to use text color other than white as it is difficult to read
Line 270 “, the grain size is bigger in the Docol 1200M than in 1000DP steel” can you please specify how much bigger it is in microns? Also can you follow up this comment with some discussion from previous literature on how does the grain size influence the weld quality, performance ..etc
The authors measured and discussed the microhardness but they didn’t mention this at all in the abstract, this must be addressed and updated.
Figure 8 is it possible for the authors to plot the graphs a and b together in one graph, this would give us a more clear understand of the hardness variation compared to each other
Figure 9 is missing the legends what does each colored line refer to?
Line 293 “smoothed using the SciDAVis software.” How much smoothing did you apply please mention that and whether it had any effects on the results?
Line 296 “exhibit a slight difference” the authors are encouraged to use more quantifying words when they describe their results as using for example the word slight is vague
“use of the short-length specimens…some distortion of information about the strain hardening behavior” why you highlight this concern, how about past studies in the literature did they report that such distortion is caused by using short specimens? Or is this just your own assumption
Lines 403-406 please support this claim by a reference/s
The authors should add a list of nomenclature for all the symbols and characters used in the manuscript, add this table at the start of manuscript just after the abstract or at the end after the conclusions
The authors provide good discussion and explanation for their findings, however in some sections especially the last few, the authors need to do the following: The authors are encouraged to include a more detailed discussion which critically discuss the observations from this investigation with existing literature.
Author Response
Dear Reviewer,
Thank you for your comments concerning our manuscript entitled “ Crash Response of Laser-Welded Energy Absorbers Made of Docol 1000DP and Docol 1200M Steels ” (Manuscript ID: materials-1209412).
Those comments are all valuable and very helpful for revising and improving our paper.
We have made a correction which we hope meet with approval. Revised portions are highlighted in yellow in the paper.
After carefully studying the Reviewer comments, we have tried our best to revise and improve the manuscript and hope that the correction of manuscript will meet with approval. Please see the attachment file.
Once again, thank you very much for your comments and suggestions.
Yours sincerely
Paweł Prochenka

Reviewer 3 Report
Please refer to the comments in the attached file.

Author Response
Dear Reviewer,
Thank you for your comments concerning our manuscript entitled “ Crash Response of Laser-Welded Energy Absorbers Made of Docol 1000DP and Docol 1200M Steels ” (Manuscript ID: materials-1209412).
Those comments are all valuable and very helpful for revising and improving our paper. We have made a correction which we hope meet with approval. Revised portions are highlighted in yellow in the paper.
Answers
- Since these laser welded steels are going to be used as the energy absorbers in the automobiles, the authors should comment on the minimum energy absorption requirement in the automobiles and do these laser welded steels meet the requirement.
Answer: The minimum absorbing energy of the vehicle's structural elements depends mainly on its design. In the article, we compare the absorption energy values mainly in the context of the assessment of the quality of the laser welded joint, but in order to be able to relate the obtained results to other literature results, we suggest supplementing the article with an additional parameter – specific energy absorbed (SEA) per unit mass.
- As shown in Figure 14b of the manuscript, the energy absorption of the Docol 1200 M laser welded steel increases with the crushing displacement. However, the Docol 1200 M steel is a martensitic steel. It demonstrates fast necking and losses the ability to bear additional load immediately after the yielding process, as presented in Figure 9b. The authors should elaborate why the energy absorption of the steel increases with increasing crushing displacement. Figure 9b and 14b of the manuscript are re-printed here for the ease of discussion. Figure 14b: The absorbed energy of Docol 1200 M during crushing under quasi-static and dynamic loadings Figure 9b: The stress-strain curves of Docol 1200 M under quasi-static and dynamic loadings Based on the discussion above, a minor revision is recommended.
Answer: Energy absorption capability is proportional to the work required to deform the absorber. Therefore, with increasing displacement crushing increases energy absorption capability. During deformation, the absorber material is subjected to complex stress state, with dominating bending stresses (effect of buckling crushing). Despite the fact that Docol 1200M steel demonstrates fast necking on stress-strain curve, no cracks in the material were notice, which were the results of a wall necking in the cross-section of the absorber.
After carefully studying the Reviewer comments, we have tried our best to revise and improve the manuscript and hope that the correction of manuscript will meet with approval.
Once again, thank you very much for your comments and suggestions.
Yours sincerely
Paweł Prochenka
Reviewer 4 Report
The article is sufficiently novel and interesting to warrant publication and it adheres to the journal's standards. The article is clearly laid out. All the key elements are present: abstract, introduction, methodology, results, discussion and conclusions. The title clearly describes the article and the abstract reflects the content of the article. The introduction contains a brief description of the actual state-of-the-art, and clearly state the problem being investigated. The authors accurately explain what they discovered in the research. The claims in conclusion are supported by the results. The references are accurate. The reviewer recommends accepting the paper for publication.
Author Response
Dear Reviewer,
The authors thank the Reviewer for the very positive assessment of our manuscript entitled “ Crash Response of Laser-Welded Energy Absorbers Made of Docol 1000DP and Docol 1200M Steels” (Manuscript ID: materials-1209412).
Yours sincerely,
Paweł Prochenka
Round 2
Reviewer 2 Report
All questions answered